# Probiotic-Induced Modulation of Microbiota Composition and Antibiotic Resistance Genes Load, an In Vitro Assessment

**DOI:** 10.3390/ijms25021003

**Published:** 2024-01-13

**Authors:** Alicja Maria Nogacka, Silvia Saturio, Guadalupe Monserrat Alvarado-Jasso, Nuria Salazar, Clara G. de los Reyes Gavilán, Ceferino Martínez-Faedo, Adolfo Suarez, Ruipeng Wang, Kenji Miyazawa, Gaku Harata, Akihito Endo, Silvia Arboleya, Miguel Gueimonde

**Affiliations:** 1Department of Microbiology and Biochemistry of Dairy Products, Instituto de Productos Lácteos de Asturias, Consejo Superior de Investigaciones Científicas (IPLA-CSIC), 33300 Villaviciosa, Spain; alicja.nogacka@ipla.csic.es (A.M.N.); silvia.saturio@ipla.csic.es (S.S.); gpemonserratjasso@gmail.com (G.M.A.-J.); nuriasg@ipla.csic.es (N.S.); greyes_gavilan@ipla.csic.es (C.G.d.l.R.G.); silvia.arboleya@ipla.csic.es (S.A.); 2Diet, Microbiota and Health Group, Instituto de Investigación Sanitaria del Principado de Asturias (DIMISA-ISPA), 33011 Oviedo, Spain; adolfo.suarez@hcabuenes.es; 3Endocrinology and Nutrition Service, Central University Hospital of Asturias (HUCA), 33011 Oviedo, Spain; ceferinofaedo@gmail.com; 4Endocrinology, Nutrition, Diabetes and Obesity Group, Institute of Health Research of the Principality of Asturias (ISPA), 33011 Oviedo, Spain; 5Digestive Service, Central University Hospital of Asturias (HUCA), 33011 Oviedo, Spain; 6Technical Research Laboratory, Takanashi Milk Products Co., Ltd., Yokohama 241-0021, Japan; z-oh@takanashi-milk.co.jp (R.W.); ke-miyazawa@takanashi-milk.co.jp (K.M.); g-harata@takanashi-milk.co.jp (G.H.); 7Department of Food, Aroma and Cosmetic Chemistry, Tokyo University of Agriculture, Abashiri, Hokkaido 099-2493, Japan; a3endou@nodai.ac.jp

**Keywords:** gut microbiota, fecal culture, probiotics, extreme obesity, antibiotic resistance genes

## Abstract

The imbalance of the gut microbiota (GM) is known as dysbiosis and is associated with disorders such as obesity. The increasing prevalence of microorganisms harboring antibiotic resistance genes (ARG) in the GM has been reported as a potential risk for spreading multi-drug-resistant pathogens. The objective of this work was the evaluation, in a fecal culture model, of different probiotics for their ability to modulate GM composition and ARG levels on two population groups, extremely obese (OB) and normal-weight (NW) subjects. Clear differences in the basal microbiota composition were observed between NW and OB donors. The microbial profile assessed by metataxonomics revealed the broader impact of probiotics on the OB microbiota composition. Also, supplementation with probiotics promoted significant reductions in the absolute levels of *tetM* and *tetO* genes. Regarding the *blaTEM* gene, a minor but significant decrease in both donor groups was detected after probiotic addition. A negative association between the abundance of Bifidobacteriaceae and the *tetM* gene was observed. Our results show the ability of some of the tested strains to modulate GM. Moreover, the results suggest the potential application of probiotics for reducing the levels of ARG, which constitutes an interesting target for the future development of probiotics.

## 1. Introduction

The gut microbiota (GM) is the group of microorganisms present in the intestinal tract. This complex and diverse microbial ecosystem is composed mainly of bacteria, also including archaea, viruses, yeast, and fungi [1]. The GM interacts with the host in a mutualistic relationship, which is critical for correct immune, metabolic, and trophic homeostasis [2]. The imbalance of the microbial composition and functionality is known as dysbiosis and is associated with different intestinal (IBS and IBD, diarrhea, celiac disease, colon cancer) and extra-intestinal conditions (neurological disorders, allergies and autoimmune diseases, metabolic syndrome, etc.) [1,3,4].

When taken together, obesity and overweight have an incidence of 39% and 20% in adults and children, respectively [5]. Obesity presents diverse symptomatology, being characterized by an excessive accumulation of body fat and low-grade systemic inflammation [6]. In addition, obesity is associated with different comorbidities [7]. Body mass index (BMI) is often used for the initial evaluation of the degree of obesity in individuals. The main BMI categories (kg/m^2^) are: normal weight (<25), overweight (25–29.9), class I obesity (30–34.9), class II obesity (35–39.9), and class III obesity or extreme obesity (≥40) [5]. Although the positive imbalance between caloric intake and energy expenditure is the main determinant of obesity, its etiology and pathogenesis are also related to other factors [8], among which the GM should also be considered [9,10,11]. The role of GM dysbiosis on the etiology of obesity has been evidenced by fecal transplant experiments, from obese or lean donors to germ-free animals, showing the transfer of the phenotype from the donor to the recipient [11,12]. These observations led to the search for defined microbiota markers in the obese phenotype [8,13,14]. Several meta-analyses in this area identified the same two recurrent characteristics in obesity: the existence of a significant and inverse relationship between microbial diversity and BMI [4,15,16,17], and the association of the obese phenotype with higher levels of fecal SCFA [8,18,19].

In spite of all this, research on extreme obesity (characterized by BMI ≥ 40 or BMI ≥ 35 plus comorbidities) and GM is still scarce, even though this condition is known to impact both quality and expectancy of life [20]. Current scientific evidence is mainly limited to studies on bariatric surgery, due to its effectiveness in weight loss and the reduction of cardiovascular risk factors [21], with few studies available in this context [22]. Although the underlying mechanisms have not been fully characterized, a low microbial diversity is strongly associated with metabolic alterations in obesity (resistance to insulin, low-grade inflammation and adipocyte hypertrophy), suggesting a relevant role of the GM [23,24]. These have drawn attention towards the development of nutritional strategies, such as the consumption of probiotics, for managing the GM in the obesity context [25].

Moreover, the increasing prevalence of microbial strains harboring antibiotic resistance genes (ARGs) is another issue of scientific interest. The GM has been reported to be a potential reservoir of ARGs. Resistances to tetracyclines and beta-lactams are particularly prevalent in the human microbiota [26] corresponding to the two antibiotic categories with larger sales in the EU [27]. Therefore, reducing the carriage of these ARGs is also of interest and a potential target for nutritional interventions. However, it is unclear to what extent the intestinal load of ARG differs among different population groups, e.g., obese vs. normal-weight subjects, and whether or not population-specific nutritional interventions may help to reduce it.

In this context, the possibility for modulating the microbiota in order to reduce the risk of disease is open [28]. The modulation of the GM can be carried out through different techniques, from the traditional use of diet to more specific manipulations through the administration of probiotic strains or prebiotic compounds. Probiotics are live microorganisms that, when administered in adequate amounts, confer a health benefit on the host [29]. Numerous clinical investigations have been undertaken into the health benefits of probiotics, with robust proven effects in several conditions [30]. 

The objective of this work was the evaluation of the abilities of different probiotic strains to modulate the microbiota composition and the levels of ARGs in two human population groups: extremely obese and normal-weight subjects. To this end we used an in vitro fecal culture model.

## 2. Results

### 2.1. Impact of Probiotics on Intestinal Microbiota Composition

The basal microbiota composition showed differences between the two studied human groups, NW and OB subjects. At the family level, significantly (*p* < 0.05) lower relative abundances of Christensenellaceae (0.00 ± 0.00% vs. 4.00 ± 2.85%, mean ± SD) and higher of Lactobacillaceae (0.68 ± 0.65% vs. 0.31 ± 0.41%, respectively) were found in the samples from OB donors. Remarkably, a preliminary comparison of the ß-diversity of microbiota communities highlighted a clear differential distribution by human donor groups in an individual-specific manner (Appendix A). 

Since clear baseline differences in the microbiota composition were found between NW and OB donors, we then studied the impacts of the different probiotic strains independently for each group of volunteers. In OB subjects’ fecal cultures, the major changes occurred in Bifidobacteriaceae and Lactobacillaceae families, with statistically significant (*p* < 0.05) increases in their relative abundance after the addition of *Bifidobacterium bifidum* TMC3115 and *Lacticaseibacillus rhamnosus* GG, respectively (Table 1). Minor, but statistically significant (*p* < 0.05), increments were also detected in the relative abundances of Rikenellaceae and Tannerellaceae families after incubation with *Bifidobacterium animalis* subsp *lactis* IPLA20020 and *Bifidobacterium longum* IPLA20022. Tannerellaceae also increased after incubation of the fecal cultures in the presence of *B. bifidum* TMC3115 and *L. rhamnosus* GG. Moreover, the strain *B. bifidum* TMC3115 was associated with a light, but significant, increase in the abundance of the Veillonellaceae, this probiotic being the one promoting more changes in the OB microbiota profile among the strains tested. Additionally, the relative abundances of several microbial groups decreased (*p* < 0.05) after incubation in the presence of probiotics: Bacteroidaceae and Ruminococcaceae families with *B. animalis* IPLA20020 and *L. rhamnosus* GG; Lachnospiraceae with *B. animalis* IPLA20020, *B. longum* IPLA20022 and *L. rhamnosus* GG, Veillonellaceae with *B. animalis* IPLA20020 and Acidaminococcaceae with *L. rhamnosus* GG. Finally, relevant decreases were observed in the Enterobacteriaceae family after incubation with all the strains tested, reaching statistical significance (*p* < 0.05) for all fecal cultures tested apart from the one added with *B. bifidum* TMC3108. Additional analyses, aiming at the comparison of changes occurring in fecal cultures among the different probiotics, highlighted the increases in the Bifidobacteriaceae family following the addition of *B. animalis* IPLA20020. Also, the Lactobacillaceae family increased more notably after *L. rhamnosus* GG addition than after the addition of any of the other strains tested (Table 1).

In fecal cultures from NW donors, probiotics presented a more limited effect on the microbiota profile (Table 2). The Bifidobacteriaceae family increased (*p* < 0.05) after the addition of *B. animalis* IPLA20020 or *L. rhamnosus* GG to fecal cultures. An increase in Lactobacillaceae was also detected after the addition of *L. rhamnosus* GG and *B. bifidum* TMC3115. Other changes observed included the increase (*p* < 0.05) promoted by *B. animalis* IPLA20020 in the Tannerellaceae family, and the slight, but significant, increase caused by *B. bifidum* TMC3115 in the Clostridiaceae1 family. Finally, the most relevant decreases were produced after incubation with *B. animalis* IPLA20020 and *L. rhamnosus* GG in the Enterobacteriaceae family. The probiotics with more prominent effects on microbiota composition in NW fecal cultures were *B. animalis* IPLA20020 for the bifidobacteria group and *L. rhamnosus* GG for the lactobacilli group. It is worth mentioning that the observed changes in NW fecal cultures were more limited than those found in fecal cultures from obese individuals where the probiotic addition caused a higher impact on the microbiota composition (Table 2). 

### 2.2. Impact of Probiotics on the Levels of Antibiotic Resistance Genes

An initial comparison of the basal ARG levels between NW and OB volunteers´ GMs indicated that none of the four ARGs analyzed presented significant differences between both human donor groups.

The levels of *tetO* and *tetM* genes decreased significantly (*p* < 0.05) as a result of the in vitro probiotic modulation of fecal microbiota from both human groups (Figure 1A–D; Appendix A). Thus, the OB and NW microbiota, after 24 h of incubation with the bifidobacteria or lactobacilli strains, presented significant reductions in the absolute levels of *tetM* and *tetO.* The only exception to this was the cultures with *B. bifidum* TMC3108, likely due to a large variability in the absolute levels obtained in the cultures with this microorganism. On the contrary, no changes in the levels of ARGs were observed in the negative control culture (no probiotic added). 

Regarding β-lactamase gene levels, we observed that probiotic addition promoted a slight but significant decrease (*p* < 0.05) in both donor groups in terms of *blaTEM* gene levels. The cultures added with the strain *B. bifidum* TMC3108 were the only ones in which the decrease did not reach statistical significance (Figure 1E,F). In contrast, the number of copies of the *blaSHV* gene remained unchanged in the OB microbiota group, whereas in the NW group the addition of *L. rhamnosus* GG and *B. bifidum* TMC3115 resulted in a significant decrease in gene copy number (Figure 1H).

The detection frequency (presence/absence) of the ARG varied between groups. NW fecal cultures showed a reduction in *tetM*, *tetO* and *blaSHV* positive cultures after incubation with the different probiotics tested. However, in the OB group the occurrence was reduced in just one of the cultures for *tetM* and in two cases for *tetO* (Figure 2). Particularly, in the NW group, fecal cultures became negative, although at a variable extent, for *tetM* after incubation with all the probiotic strains assayed, whereas *B. animalis* IPLA20020 and *B. bifidum* TMC3108 promoted undetectable levels of *tetO* gene in some cultures. An important reduction in the frequency of cultures positive for the *blaSHV* gene was also seen after incubation with *B. bifidum* TMC3115 and TMC3108, *B. longum* IPLA20022 and *L. rhamnosus* GG. The ARG levels in cultures of the OB group only became undetectable in specific cases: when *B. bifidum* TMC3108 was added, it led to a clear reduction in cultures positive for *tetM,* and a certain reduction was obtained with *B. bifidum* TMC3115 and *L. rhamnosus* GG in cultures positive for *tetO*. The only gene whose occurrence was not reduced after the addition of any of the probiotics tested was *blaTEM*, neither in OB nor in NW fecal cultures. 

### 2.3. Associations between ARGs and Probiotic-Modulated Microbiota 

With the aim of deepening the associations between variations in antibiotic burden and in microbiota composition, Spearman correlations were determined independently for OB and NW fecal cultures according to the differences in the basal composition and the observed effects of probiotics (Figure 3).

In the OB group, the strongest correlation coefficient found was a positive association between the levels of *tetO* and the relative abundance of the family Clostridiaceae1, as well as a negative association with Erysipelotrichaceae and Desulfovibrionaceae. In the NW group, *tetO*, *tetM* and *blaTEM* showed a similar trend, suggesting a similar association between changes in the microbiota composition and the levels of these ARG (Figure 3). The most relevant associations detected were the positive correlation between the *tetO* levels and the Enterobacteriaceae family, and between Christensenellaceae abundance and the levels of *tetM, tetO* and *blaTEM* genes. The ß-lactamase gene *blaTEM* showed an inverse association with the presence of Bacteroidaceae and Lachnospiraceae families.

After observing these correlations between microbial taxa and ARG levels, we conducted a stepwise regression analysis to formulate a model that incorporates multiple predictors and adjusting by individual (Appendix A). Among the different regression models where probiotic addition could be the major predictor, we found a remarkable negative association between the Bifidobacteriaceae family and the *tetM* gene in both human groups. In the OB group, this negative coefficient was higher and presented a unique predictor (β −0.57, *p*-value 0.00), while in NW the association was lower, and multiple family taxa composed it (β −0.28, *p*-value 0.04). The representation of Spearman correlation between the found *tetM* gene and bifidobacteria enabled us to observe lower levels of ARGs after the addition of bifidobacteria, particularly in the OB cohort (Figure 4).

## 3. Discussion

The composition of the basal GM influences the host´s response to intervention with diet, drugs, or functional supplements. Taking this into consideration, we demonstrated clear differences between the two human groups analyzed in the present study, NW and OB donors. The comparison between both groups revealed different microbiota profiles, with lower Christensenellaceae and higher Lactobacillaceae levels in the OB group, which is in agreement with previous reports [22]. Differences between human groups, as well as interindividual differences, in the GM could be determinant for the fate of an administered probiotic strain [31]. This underlines the need for the selection of specific probiotics for well-defined target populations by analyzing the impact of the probiotics tested in a human group-specific manner. Currently, probiotic strains are evaluated based on their safety and efficacy, following the recommendations of the FAO/WHO [32], and they must prove their efficacy in human intervention studies [33]. This last requirement is challenging, mainly due to the cost of clinical trials and the difficulties in the evaluation of beneficial aspects associated to health. Thus, cheap and easily performed tests with the ability to consider the influence of the basal microbiota, such as fecal cultures, are often used for the preliminary screening and selection of strains. 

In our fecal culture model, the addition of probiotics promoted changes in the microbiota composition, mostly in the bifidobacteria and lactobacilli groups, taxa to which the probiotics tested belong, although variations in other microbial groups were observed as well. In fecal cultures from OB subjects, the strains *B. animalis* IPLA20020, *B. bifidum* TMC3115 and *B. bifidum* TMC3108 promoted major increases in the relative abundance of the Bifidobacteriaceae family, in line with previous observations [34]. These changes were accompanied by decreases in other microbial groups such as Lachnospiraceae, Ruminococcaceae and Enterobacteriaceae families. In the same way, a minor decrease in Bacteroidaceae was detected after the addition of *B. animalis* IPLA20020 or *L. rhamnosus* GG to the fecal cultures. The literature on the impact of probiotics in the GM of extremely obese individuals is still scarce and mainly limited to the effect of probiotics in bariatric surgery [35,36]. In the present work, it is remarkable that deeper changes were obtained by the addition of probiotics to the fecal cultures of the OB group as compared to those of the NW group, which is especially relevant regarding the effect of the strains tested on the increase in the Bifidobacteriaceae family. A similar differential effect was previously reported by us after in vitro modulation of the fecal microbiota of OB and NW individuals with prebiotic compounds [37], suggesting that the microbiota of extremely obese subjects is more unstable and prone to changes compared to the microbiota of healthy subjects.

In addition to these general changes in GM composition, we evaluated the impact of probiotics on the levels of ARGs as an interesting quantifiable trait. Since probiotics have been repeatedly reported to modulate the GM composition, there is a possibility that these microorganisms also affect the ARG repertoire and levels within the gut. For this reason we assessed the impact of the tested strains on the numbers of copies of some defined ARGs. Resistance to tetracycline, a family of broad-spectrum antibiotics, can be evaluated by several techniques and based on the molecular mechanism conferring resistance. Herein, we have focused on ribosomal protection genes, which are the most common in the human gut [38]. A preliminary screening was performed on the frequency detection of ARG-positive fecal cultures after probiotic addition. The majority of the fecal cultures remained ARG-positive after probiotic addition, especially regarding the *blaTEM* gene; however, in the case of *blaSHV*, *tetO* and *tetM,* some probiotics led to decreases in ARG frequency, turning them undetectable. The most prominent reductions in tetracycline resistance genes were observed after *B. bifidum* (TMC3115 and TMC3108) and *L. rhamnosus* GG addition. Looking more in detail at the absolute levels of the assessed ARG, a broad decrease in copy numbers of the two genes codifying for tetracycline resistance was detected in both human groups after probiotic addition. Similar results have been previously described in an intervention study with commercially available probiotic supplementation, although the study reported that the beneficial effect was restricted to the individuals permissive to probiotic colonization [39]. Our data suggest the potential of probiotics for modulating the intestinal reservoir of ARG, although this in vitro evidence should be corroborated in vivo. Moreover, our data do not provide mechanistic insight beyond associations. It should also be considered that, in addition to the effect on the number of copies of a certain ARG, probiotics may also have an impact on gene expression in the intestine, although such an aspect was not addressed here.

Nowadays, some modern techniques allow for linking an antibiotic resistance gene to its bacterial host [40]. With the aim of understanding the link between ARGs and probiotic-modulated GM composition, we conducted correlation analyses followed by stepwise regression models. In first place, in OB fecal cultures, we found a positive relationship between levels of the *tetO* gene and Clostridiaceae, which could be related to the greater presence of genes conferring resistance to tetracycline detected primarily in taxa related with the family Clostridiaceae [41]. Regarding NW fecal cultures, associations between ARG levels (*tetO*, *tetM* and *blaTEM* genes) and changes in microbiota composition triggered by probiotic addition were also observed. Among them, the strongest association (correlation coefficient 0.62; *p*-value 0.000) was found between *tetO* and the Enterobacteriaceae family, in line with a previous study reporting an enrichment of this microbial group after antibiotic treatment and the inherent resistance to tetracycline of microorganisms from this family [42]. The second most important association observed was the positive correlation between several ARGs and Christensenellaceae abundance, which could be related to the contents of ARGs in the genomes of members of this family [43,44]. Finally, in the context of probiotic addition, the stepwise regression model revealed a negative association between the relative abundance of bifidobacteria and the *tetM* gene, especially in the OB group. This association confirms that the addition of probiotics, mainly from the *Bifidobacterium* genus, is associated with a decrease in the levels of this gene in the microbiota. The impact of probiotics on microbiota and ARG levels highlights the importance of considering the ecological context in which probiotics are supplemented. 

To sum up, our results show the ability of some of the tested strains to modulate the microbiota in NW and OB subjects. Moreover, the results suggest the potential application of probiotics for reducing the ARG load in the human GM, which constitutes an interesting target for the rational development of population-specific probiotic products.

## 4. Materials and Methods

### 4.1. Fecal and Probiotic Culture Conditions 

A fecal batch culture methodology described previously [34] was used for the study of the impact of probiotic strains. Briefly, independent pH-free batch fermentations were performed, by duplicate, with fecal samples of individuals from two population groups, namely, normal-weight (NW, *n* = 9) and extremely obese (OB, *n* = 6) human donors. NW (BMI < 25 kg/m^2^) and OB volunteers (BMI ≥ 40 kg/m^2^), with mean ages 40 ± 9 and 44 ± 9 years, respectively, were recruited at the Digestive and Endocrinology and Nutrition Services of the Asturias Central University Hospital (HUCA, Asturias, Spain). All participants followed an unrestricted diet and had not taken antibiotics during the previous 6 months and probiotics or prebiotics during the previous 2 months. The study was approved by the Bioethical Committee of CSIC and by the Regional Ethics Committee for Clinical Research of the Principality of Asturias (Ref 13/2007, 11/03/07) in compliance with the Declaration of Helsinki of 1964, last revised in 2013. Informed written consent was obtained from each volunteer. Samples were collected and immediately introduced into anaerobic jars (Anaerocult A System, Merck, Darmstadt, Germany) for transportation to the laboratory within 1 h and stored at −80 °C until use. Fecal samples were thawed under anaerobic conditions, diluted 1/10 (*w*/*v*) in PBS, homogenized in a Lab-Blender 400 stomacher (Seward Medical, London, UK) and used as inocula for the fecal culture experiments in a carbohydrate-free basal medium [45]. After an overnight period of microbiota stabilization at 37 °C under anaerobic conditions, the cultures were added with a carbon source (0.3% (*w*/*v*) of a fructooligosaccharide (1-kestose, β-Food Science Co., Ltd., Tokyo, Japan)) and each one of the five probiotic strains was added to a culture vessel. Bacterial strains were inoculated to fecal cultures at a final concentration of 1 × 10^8^ CFU/mL. A total of four *Bifidobacterium* strains (*Bifidobacterium animalis* subsp *lactis* IPLA20020, *B. bifidum* TMC3108, *B. bifidum* TMC3115 and *B. longum* IPLA20022) and an *L. rhamnosus* (formerly *Lactobacillus rhamnosus* GG (ATCC53103)) were tested separately. The culture conditions of probiotic strains prior to the addition to fecal cultures were described previously [45]. Inoculated fecal cultures were then incubated under anaerobic conditions at 37 °C for 24 h. Samples were taken at time 0 before incubation (basal conditions) and after 24 h of incubation. These samples were centrifuged at full speed for 10 min and cell pellets were stored at −20 °C until further analyses.

.

### 4.2. Microbiota Composition by Metataxonomic Analyses

The microbial taxonomic composition of pellets from fecal batch cultures was assessed by 16S rRNA gene sequencing of the V3–V4 region. In brief, the QIAamp DNA Stool Mini Kit (Qiagen; Düsseldorf, Germany) was used for genomic DNA extraction as per the manufacturer’s protocol. The DNA concentration was analyzed with a Qubit 3.0 fluorometer (Thermo Fisher Scientific, Whaltham, MA, USA). 16S libraries were constructed according to the Illumina-recommended “16S Metagenomic Sequencing Library Preparation” protocol. 

In detail, DNA PCR amplification was performed with a TaKaRa PCR Thermal Cycler Dice Touch (Takata Bio Inc, Shiga, Japan) and 2 × KAPA HiFi HotStart ReadyMix (Kapa Biosystems, Wilmington, MA, USA) under the following conditions: initial denaturation at 95 °C for 3 min, followed by 25 cycles of 95 °C for 30 s, 55 °C for 30 s, and 72 °C for 30 s, and final extension step at 72 °C for 5 min. Indexed PCR was performed using a Nextera XT Index Kit v2 (Illumina, San Diego, CA, USA) according to the manufacturer’s directions. PCR products were purified with AMPure XP magnetic beads (Beckman Coulter Inc., Carlsbad, CA, USA) in order to remove primer dimers, and the integrity was assessed through a 1.0% (*w*/*v*) agarose gel electrophoresis. Next, 5 μL of each diluted DNA amplicon solution was mixed to prepare the pooled final library. The DNA concentration of the pooled final library was estimated using a Qubit 3.0 fluorometer, diluted to 4 nM. Sequencing was performed using an Illumina MiSeq sequencer with MiSeq Reagent Kit v3 chemicals. 

For the microbial sequence analysis, the assembled reads were processed using the QIIME2 platform (v. 2019.10). Sequence reads were imported into QIIME2 with quality assessment, filtering, barcode trimming, and chimera detection performed using the DADA2 pipeline. A taxonomic classification was assigned to amplicon sequence variants using the SILVA release 132 with taxonomic classification at >99% confidence.

### 4.3. Quantification of Antibiotic Resistance Genes 

Quantification of the fecal levels of four genes encoding resistance to two different antibiotic families, β-lactams (*bla*_TEM_, *bla*_SHV_) and tetracyclines (*tet*O, *tet*M), was achieved using qPCR. All genes were amplified on MicroAmp optical plates sealed with MicroAmp optical caps (Applied Biosystems, Foster City, CA, USA) in a 7500 Fast Real Time PCR System (Applied Biosystems) using primers and conditions detailed in Table 1. Reactions were carried out in a total volume of 25 µL, containing 1 µL of template fecal DNA (~5 ng) and 0.2 µM of each primer and 2× SYBR Green PCR Master Mix (Applied Biosystems). Thermal cycling consisted of an initial cycle of 50 °C for 5 min and 95 °C for 10 min, followed by 40 cycles at 95 °C for 15 s, and 1 min at the appropriate primer pair-annealing temperature (Table 3), followed by a dissociation curve to verify the specificity of the amplified products. Electrophoresis runs and sequencing were also performed to confirm the specificity of the PCR products. Standard curves were made with known quantities of cloned target genes. In brief, target genes were amplified by conventional PCR (UnoCycler, VWR International, West Chester, PE, USA) using primers described in Table 2 from bacterial DNA of pure cultures. PCR products were visualized in agarose gels by electrophoresis, purified using a PCR Purification Kit (Qiagen Inc., Valencia, FL, USA) and sequenced for verification. Amplicons were subsequently cloned into a pGEM^®^-T Easy vector (Promega, Madison, WI, USA), and transformed into *Escherichia coli* JM109 competent cells following the manufacturer´s indications. Plasmids were isolated using a High Pure Plasmid Isolations kit (Sigma-Aldrich, San Luis, MO, USA), and the concentration was determined in a NanoDrop apparatus (Implen GmbH, Munich, Germany); gene copy numbers (GCN) were then calculated as described elsewhere [46]. Tenfold serial dilutions of DNA vector were used as the standard curve for qPCR. Amplifications by qPCRs were carried out with high efficiencies for each gene (96.92 ± 3.82) and the limit of quantification was 3.56, 4.42, 3.93 and 4.36 log_10_ no. copies/mL for *bla*_TEM_, *bla*_SHV_, *tet*O and *tet*M, respectively. All samples were analyzed in duplicate in at least two independent PCR runs and the results are presented as log_10_ gene copies/mL calculated as the mean of two replicates. 

### 4.4. Statistical Analyses

All experimental data are reported as mean ± standard deviation. The statistical analysis of results was performed using the software SPSS v.26 (SPSS Inc., Chicago, IL, USA). The basal microbiota composition was compared between human groups by the non-parametric U-Mann–Whitney test. ß-diversity analysis was carried out using QIIME 2 through a Principal Coordinate Analysis (PCoA) of unweighted UniFrac distances plotted using Emperor. The impacts of different probiotic strains on microbiota composition and antibiotic resistance genes were studied by analyzing changes in time by the Wilcoxon signed rank test, whereas Kruskal–Wallis tests were conducted for changes among the evaluated probiotics. The putative associations between microbiota groups, at the family taxonomic level, and the ARGs, as modulated by the probiotics strains, were evaluated by Spearman correlation and plotted in the RStudio version 1.2.5001 by package rcorr and corrplot command, independently for NW and OB fecal cultures. Significant correlations were introduced in a stepwise regression analysis between the microbial families and ARG levels, adjusting by individual.

## Figures and Tables

**Figure 1 ijms-25-01003-f001:**
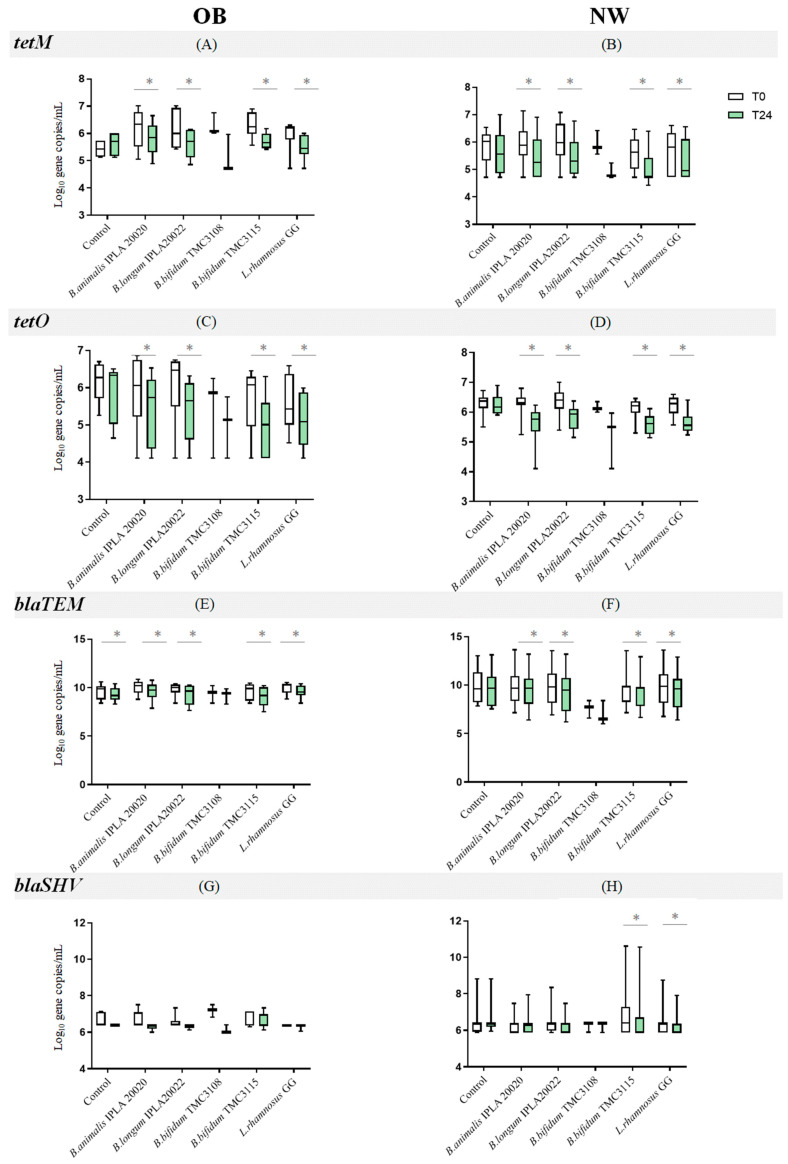
Levels of antibiotic resistance genes *tetM*, *tetO*, *blaTEM* and *blaSHV* (Log_10_ gene copies/mL) in OB (**A**,**C**,**E**,**G**) and NW (**B**,**D**,**F**,**H**) fecal cultures before (white boxes) and after (green boxes) probiotics modulation. * denotes statistically significant differences (*p* < 0.05).

**Figure 2 ijms-25-01003-f002:**
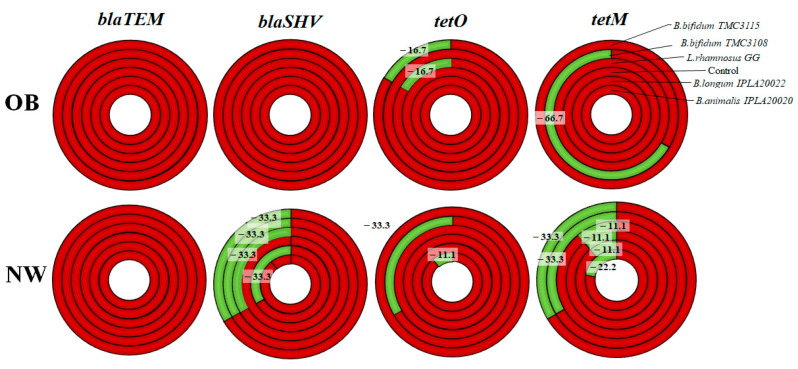
Detection frequency of ARG presence in fecal cultures after probiotic addition in OB and NW fecal cultures. The red color represents the frequency of ARG-positive fecal cultures, and green represents the frequency reduction and the apparition of ARG-negative fecal cultures.

**Figure 3 ijms-25-01003-f003:**
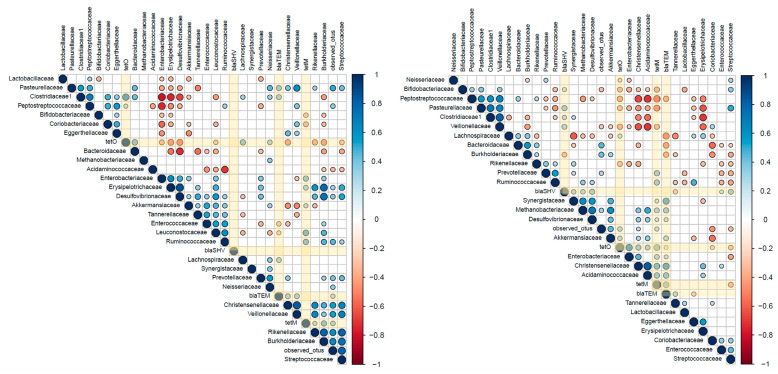
The Spearman correlations among ARGs and gut bacterial families in fecal cultures from normal-weight (right) and obese (left) adults after probiotic addition. Blue and red colors denote positive and negative associations, respectively. The intensity of the colors and the areas of circles show the absolute values of corresponding correlation coefficients. Only significant associations (*p* < 0.05) are represented.

**Figure 4 ijms-25-01003-f004:**
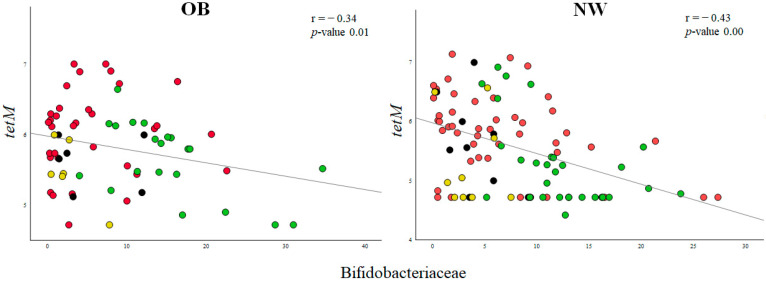
Correlations between *tetM* gene levels (Log10 gene copies/mL) and abundance of Bifidobacteriaceae family (%) in OB and NW fecal cultures at time 0 (red circles) and after the addition of bifidobacteria strains (green circles) and lactobacilli (yellow circle), and in controls without probiotic addition (grey circle). r and *p*-values are given following the Spearman correlation test.

**Table 1 ijms-25-01003-t001:** Changes over time (T24-T0) in the microbiota composition (relative abundance, mean ± SD) at the family–taxonomic level in OB fecal cultures after incubation over 24 h with different probiotic strains. Significant results of the Wilcoxon signed rank test are represented with “*” and “**” for *p*-value < 0.05 and <0.01, respectively. The statistically significant differences (*p*-value < 0.05) obtained on the Krustal-Wallist test when comparing the cultures added with the different strains, are indicated by different superscript letters.

Family Level	Control	*B. animalis* IPLA20020	*B. bifidum* TMC3108	*B. bifidum* TMC3115	*B. longum* IPLA20022	*L. rhamnosus* GG
Bifidobacteriaceae	4.47 ± 4.62 ^ab^	11.00 ± 2.92 ^b^	8.09 ± 5.65 ^ab^	8.03 ± 10.87 *^ab^	2.08 ± 6.76 ^a^	1.75 ± 1.83 ^a^
Coriobacteriaceae	1.32 ± 1.04 *	0.39 ± 1.25	0.89 ± 0.96	−0.35 ± 1.38	−0.04 ± 1.28	−0.35 ± 2.16
Bacteroidaceae	−0.89 ± 5.99	−1.33 ± 2.71 *	−0.59 ± 1.54	−1.17 ± 2.97	1.02 ± 3.83	−2.75 ± 4.24 *
Rikenellaceae	1.19 ± 2.19	0.33 ± 0.89 **	0.42 ± 0.73	0.20 ± 0.49	0.29 ± 0.57 *	0.04 ± 0.33
Tannerellaceae	6.48 ± 3.67	3.33 ± 3.41 **	1.83 ± 3.40	1.50 ± 3.07 **	1.30 ± 2.97 *	1.46 ± 3.51 *
Lactobacillaceae	2.41 ± 6.08 ^a^	1.30 ± 3.93 ^a^	0.49 ± 0.17 ^ab^	2.99 ± 4.93 ^a^	2.05 ± 4.79 ^a^	19.34 ± 8.99 **^b^
Streptococcaceae	−0.16 ± 0.96	−0.79 ± 1.14	0.18 ± 0.88	−0.50 ± 1.00	0.11 ± 0.53	−0.28 ± 0.28
Christensenellaceae	−0.28 ± 0.26 *	−0.20 ± 0.33	−0.09 ± 0.28	−0.11 ± 0.09	−0.08 ± 0.16	−0.11 ± 0.19
Clostridiaceae	−3.20 ± 3.47	−2.40 ± 3.09	−1.94 ± 2.94	−2.69 ± 2.34	−3.19 ± 3.70	−3.10 ± 3.30
Lachnospiraceae	−2.75 ± 6.46	−3.76 ± 6.14 *	−2.86 ± 2.17	−2.03 ± 7.08	−2.40 ± 7.25 *	−3.94 ± 5.00 *
Ruminococcaceae	−1.82 ± 4.37	−4.82 ± 5.67 **	−4.61 ± 0.66	−3.56 ± 4.53	−1.29 ± 4.86	−5.01 ± 5.81 *
Erysipelotrichaceae	1.17 ± 4.83 *	0.64 ± 3.18	−0.92 ± 1.01	0.12 ± 1.12	0.37 ± 1.51	−0.43 ± 1.85
Acidaminococcaceae	1.68 ± 3.74	3.32 ± 4.05	2.59 ± 4.56	3.74 ± 4.70	1.36 ± 1.49	−1.04 ± 3.33 *
Veillonellaceae	−0.47 ± 1.96	−0.28 ± 0.74 *	1.27 ± 1.11	0.06 ± 0.92 *	−0.13 ± 0.34	−0.29 ± 1.05
Burkholderiaceae	−0.84 ± 0.94	−1.12 ± 1.15	−0.46 ± 0.58	−0.82 ± 1.20	−0.67 ± 0.86	−1.04 ± 1.09
Enterobacteriaceae	−6.28 ± 3.73 *	−3.68 ± 1.90 **	−3.55 ± 3.85	−3.21 ± 4.71 **	−1.54 ± 3.55 *	−4.37 ± 3.66 **

**Table 2 ijms-25-01003-t002:** Changes over time (T24-T0) on the microbiota composition (relative abundance, mean ± SD) at the family–taxonomic level in NW fecal cultures after incubation over 24 h with different probiotic strains. Significant results of the Wilcoxon signed rank test are represented with “*” for *p*-value < 0.05, respectively. The statistically significant differences (*p*-value < 0.05) obtained on the Krustal-Wallist test when comparing the cultures added with the different strains, are indicated by different superscript letters.

Family Level	Control	*B. animalis* IPLA20020	*B. bifidum* TMC3108	*B. bifidum* TMC3115	*B. longum* IPLA20022	*L. rhamnosus* GG
Bifidobacteriaceae	1.66 ± 2.54 *	4.19 ± 8.01 *	6.55 ± 4.62	4.26 ± 4.90	0.31 ± 5.90	1.32 ± 2.48 *
Coriobacteriaceae	3.30 ± 4.17	0.78 ± 1.14	0.18 ± 0.31	0.61 ± 1.70	0.62 ± 1.48	2.37 ± 2.65
Bacteroidaceae	−2.02 ± 6.83	−1.66 ± 1.74	−1.59 ± 1.80	0.74 ± 3.93	2.86 ± 5.43	−2.78 ± 3.19
Rikenellaceae	0.62 ± 0.89	0.84 ± 1.31	0.06 ± 0.72	0.22 ± 0.56	0.54 ± 0.62	0.11 ± 0.70
Tannerellaceae	3.99 ± 4.57 *	4.95 ± 2.88 *	2.35 ± 0.30	3.07 ± 2.43	5.41 ± 6.92	2.09 ± 2.21
Lactobacillaceae	0.04 ± 0.12 ^a^	0.08 ± 0.18 ^a^	0.01 ± 0.01 ^ab^	0.05 ± 0.15 *^a^	0.19 ± 0.31 ^a^	12.89 ± 5.86 *^b^
Streptococcaceae	−1.55 ± 6.12	−0.25 ± 2.68	0.55 ± 0.72	−0.39 ± 1.95	0.30 ± 0.88	−0.53 ± 1.64 *
Christensenellaceae	−1.71 ± 1.88 *	−0.12 ± 1.15	−0.14 ± 0.88	−0.65 ± 0.94 *	0.14 ± 1.54	−1.01 ± 1.74
Clostridiaceae	0.39 ± 1.04 *	0.90 ± 3.17	−0.15 ± 0.15	0.79 ± 2.63 *	1.52 ± 4.76	−0.31 ± 0.60 *
Lachnospiraceae	−0.92 ± 4.62	−2.91 ± 2.72	−1.15 ± 3.41	−2.12 ± 3.28	−2.04 ± 3.69	−3.24 ± 2.75 *
Ruminococcaceae	2.27 ± 10.63	−4.02 ± 3.85	−3.08 ± 3.51	−2.12 ± 3.92	−0.98 ± 6.77	−5.06 ± 4.35
Erysipelotrichaceae	1.31 ± 1.74	2.03 ± 2.70	0.78 ± 1.95	1.10 ± 1.78	1.38 ± 2.33	0.73 ± 1.68
Acidaminococcaceae	−1.17 ± 2.08	−0.42 ± 1.59	−0.58 ± 0.99	−0.52 ± 1.02	−0.54 ± 1.38 *	−1.65 ± 1.89
Veillonellaceae	0.35 ± 0.77	0.34 ± 0.68	0.14 ± 0.17	0.37 ± 0.65	0.43 ± 0.63	0.36 ± 0.98
Burkholderiaceae	0.91 ± 1.76	0.49 ± 1.04 *	−0.14 ± 0.32	0.12 ± 0.74	0.31 ± 0.62 *	−0.14 ± 0.89 *
Enterobacteriaceae	−5.78 ± 7.55 *	−4.49 ± 3.00 *	−2.25 ± 2.89	−4.40 ± 3.12	0.33 ± 9.71	−4.13 ± 3.59 *

**Table 3 ijms-25-01003-t003:** Assessment of the fecal levels of four genes encoding resistance to two different antibiotic families, β-lactams (*blaTEM*, *blaSHV*) and tetracyclines (*tetO*, *tetM*). “Tª” annealing temperature; “REF” reference; “ARG” antibiotic resistance genes.

ARG	Gene	Primer	Sequence 5′–3′	Size (pb)	Tª (°C)	Ref
β-lactams	*blaSHV*	bla-SHV F PCR	CACTCAAGGATGTATTGTG	885	58	[47]
		bla-SHV R PCR	TTAGCGTTGCCAGTGCTCG			
		bla-SHV F qPCR	GCTGGAGCGAAAGATCCACT	247	60	[48]
		bla-SHV R qPCR	CGCCTCATTCAGTTCCGTTT			
	*bla_TEM_*	bla-TEM F	CACTATTCTCAGAATGACTTGGT	85	60	[49]
		bla-TEM R	TGCATAATTCTCTTACTGTCATG			
Tetracyclines	*tetO*	tetO F	ACGGARAGTTTATTGTATACC	171	60	[50]
		tetO R	TGGCGTATCTATAATGTTGAC			
	*tetM*	tetM F	ACAGAAAGCTTATTATATAAC	171	55	[50]
		tetM R	TGGCGTGTCTATGATGTTCAC			

## Data Availability

Additional data presented in this study are available upon reasonable request from the corresponding author.

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
