# Peer review of "Probiotic-Induced Modulation of Microbiota Composition and Antibiotic Resistance Genes Load, an In Vitro Assessment"

_ijms, 2024, doi:10.3390/ijms25021003_

Round 1
Reviewer 1 Report
Comments and Suggestions for Authors
The study presents intriguing findings on how probiotics impact gut microbiota (GM) composition and antibiotic resistance genes (ARG) in obese and normal-weight individuals. However, the methods and data analysis need more detail for clarity. The link between probiotics, GM modulation, and ARG reduction is promising but requires further exploration. Additionally, consider expanding on the mechanism behind the observed negative association between Bifidobacteriaceae and tet genes. A more thorough examination of these aspects would strengthen the study.
Minor revision recommended.
1. Line 157: The three-line table format in the article needs adjustment.
2. Line 249: The clarity of Figure 3 in the article needs to be improved.
3. Line 254: In the discussion section, it is recommended that the authors should focus on discussing why these probiotics can influence the expression of resistance genes. What are the potential applications?
4. Line 249: The p-value notation in Figure 3, 'p-value 0.00', should be standardized.
Comments on the Quality of English Language
The study presents intriguing findings on how probiotics impact gut microbiota (GM) composition and antibiotic resistance genes (ARG) in obese and normal-weight individuals. However, the methods and data analysis need more detail for clarity. The link between probiotics, GM modulation, and ARG reduction is promising but requires further exploration. Additionally, consider expanding on the mechanism behind the observed negative association between Bifidobacteriaceae and tet genes. A more thorough examination of these aspects would strengthen the study.
Minor revision recommended.
1. Line 157: The three-line table format in the article needs adjustment.
2. Line 249: The clarity of Figure 3 in the article needs to be improved.
3. Line 254: In the discussion section, it is recommended that the authors should focus on discussing why these probiotics can influence the expression of resistance genes. What are the potential applications?
4. Line 249: The p-value notation in Figure 3, 'p-value 0.00', should be standardized.
Author Response
We thank the reviewer for his assessment of our manuscript. We have trid to answer the points raised as follows;
- Line 157: The three-line table format in the article needs adjustment.
We are not sure which specific aspects of the format the referee refers to. We have slightly modified the table format, adjusting columns width and adding a line on top of the table to make it clearly different from the text. Since table 1 and 2 had the same format we have also modified the format of table 2.
- Line 249: The clarity of Figure 3 in the article needs to be improved.
Figure 3 is now replaced by a higher resolution version and renumbered as Figure 4 dude to the inclusion of an additional figure as suggested by one of the reviewers.
- Line 254: In the discussion section, it is recommended that the authors should focus on discussing why these probiotics can influence the expression of resistance genes. What are the potential applications?
This part has been developed further in the discussion section, also acknowledging some existing gaps, such as the potential impact of probiotics on ARG expression, not just on ARG copies, that are not addressed in our study. To this end the following texts were added;
“Since probiotics have been repeatedly reported to modulate the GM composition there is a possibility for these microorganisms to also affect the ARG repertoire and levels within the gut. For this reason we assessed the impact on the number of copies of some defined ARG in the fecal samples” (Lines 295-298).
And
“Our data suggest the potential of probiotics for modulating the intestinal reservoir of ARG, although the data should be corroborated in vivo. Moreover, our data do not provide mechanistic insight beyond associations. It should also be considered that, in addition to the effect on the number of copies of a certain ARG, probiotics may also have an impact on gene expression in the intestine, although such aspect was not addressed here.” (Lines 315-320).
Other than these some minor changes have been done in the discussion section to slightly reduce other parts, perhaps less relevant, of it in order to render a discussion most focused on the ARG, as the reviewer suggests.
- Line 249: The p-value notation in Figure 3, 'p-value 0.00', should be standardized.
This is standardized since the values given correspond to the actual p-values (p=0.01 in one case and p=0.00 in the other)
Reviewer 2 Report
Comments and Suggestions for Authors
This paper aimed to explore probiotic-induced modulation of microbiota composition and antibiotic resistance genes load, which was of general interest to IJMS. Overall, the research is meaningful. However, the paper should be improved, previously to publication. I have listed some comments below:
1. Line 344. Why are there different numbers of people in the NW group and the OB group?
2. Line 351. Why was a bacterial strain concentration of 1*108 cfu/mL chosen for inoculation into the fecal culture?
3. Lines 351-354. Why were these four specific strains of Bifidobacterium and this specific strain of Lactobacillus rhamnosus chosen for the experiment.
4. Please update the references to evaluate the latest research progress in this field.
5. Figure 1 and 3. Please ensure consistency in the fonts used in both the images and their captions.
6. Minor editing of English language required
Comments on the Quality of English LanguageMinor editing of English language required
Author Response
We thank the reviewer for the constructive assessment of our manuscript. We have tried to answer the points raised as follows;
- Line 344. Why are there different numbers of people in the NW group and the OB group?
Our initial aim was 6 donor per group, since n=6 is commonly used in this sort of studies and we have use this number before. However, when recruiting healthy controls our previous experience indicate that in this group there are more failures, since these individuals do not feel a direct benefit of participating in the study and the faeces collection and transportation to the hospital is not a handy procedure. For this reason we contacted an extra number of subjects from this group. Then, these 9 volunteers accepted to participate and provided the sample and we felt it would be unethical not using them which, in addition, increased the sample size in that groups which is always advantageous.
- Line 351. Why was a bacterial strain concentration of 1*108cfu/mL chosen for inoculation into the fecal culture?
This is always an issue open to discussion, since some studies use higher numbers. In our case we decided this concentration trying to get a plausible physiological situation. When consuming a probiotic product in most cases numbers between 10E10 and 10E11 are consumed (100 ml of a product with 10E8-10E9 ufc/mL). With survivals through GIT transit been often limited (<10% and very often even lower) we though it would be not too realistic using concentrations higher that 10E8-10E9, and we decided to be in the lower part of that range. Moreover, these are concentrations that are commonly present for these microorganisms in human faeces. This promted us to use the concentration of 10E8, trying to be a bit conservative to avoid large overestimation of the potential effects due the introduction in the in vitro system of excessively large numbers of the strains to be tested.
- Lines 351-354. Why were these four specific strains of Bifidobacteriumand this specific strain of Lactobacillus rhamnosus chosen for the experiment.
Of course these represent just a limited number of strains and many other strains could have been tested in the system. However, this study was pioneer in trying to select probiotics for reducing ARG load in defined human populations and, in this sense, constituted a “proof of concept”, setting the methods for later testing other strains and/or compounds. Thus, with this in ming we had to select a limited number of strains for this initial testing and we had an ongoing collaboration with a dairy company owner of two of the strains tested (as acknowledged in the manuscript) who compromised to partially cover the expenses of the study. Then we included the strain that, likely, represent the most studied probiotic lactobacilli (LGG) and two strains from our own collection that we had previously characterized extensively (among these a B. lactis strain is included since B. lactis is the most commonly used probiotic bifidobacterial species in foods).
- Please update the references to evaluate the latest research progress in this field.
We are not sure which references the referee refer as out of date. We have tried to use references that we think still hold valid for the statements they give support to. Then, in the specific setting of the scope of this study there are some aspects, such as the impact of probiotics on ARG where the references available are still very limited.
- Figure 1 and 3 (figure 3 is now renumbered as Figure 4 after addition of an additional Figure as suggested by one of the reviewers). Please ensure consistency in the fonts used in both the images and their captions.
These have been modified for consistency. In the case of Figure 3 also for increased resolution as requested by another reviewer.
- Minor editing of English language required
The manuscript has been proof read and some minor modifications have been done.
Reviewer 3 Report
Comments and Suggestions for Authors
In the article "Probiotic-induced modulation of microbiota composition and antibiotic resistance genes load, an in vitro assessment", the authors have used an in vitro set up to test the phenotypic effects of specific probiotics on the gut microbiota grown out from the fecal material from either obese patients (OB) or normal weight patients (NW). Their findings show that OB microbiota differ from NW microbiota, and addition of probiotics influences overall community structure (more dramatically in OB microbes) as well as amount of antibiotic resistance genes (ARG). This article has some interesting ideas, but I would like to see some additional experiments as well as the data displayed in a different manner for clarity. Overall, the premise suggests the potential role for probiotics as a method of actively influencing dysbiotic microbiomes associated with obesity as well as influencing the potential for transfer of antibiotic resistance. My suggestions are as follows:
1. as the fecal material was obtained from patients, is there any information as to whether these individuals had a consistent diet supplementation of probiotics or probiotic/fermented foods? If this is unknown, a brief discussion of how this might have influenced the individual microbiota should be included as a potential confounding factor.
2. Perhaps my biggest issue comes from the lack of characterization of what might be outgrowth of the added probiotic bacteria vs the bacteria from the OB/NW microbiota. Given that 10^8 probiotic are being added to the cultures, and these organisms also have the ability to process the nutrients in the culture (perhaps even more readily than the fecal microbes), how does one know that the substantial increases in both Bifido and Lacto are not actually the probiotic expanding. This consideration also applies to the measure of number of ARGs. How do the authors know that the decrease in ARGs is not actually just do to outcompetition in the culture by the added probiotic? Or, that the probiotics themselves are possibly contributing genes?
3. This brings me to Figure 1. There are several issues I have with this figure.
a. The y-axis needs to be labeled or described in the legend as it is not intuitive.
b. I have a very hard time believing that statistical significance was found between most of these samples with the broad distributions and the size of the error bars. Please include an associated table with the actual numbers included so the author can see where the significance comes from.
c. What is happening with the B. bifidum TMC3108 samples? The loss of the distribution for every single one of these samples at the very least needs some further explanation.
d. I don't think this is the best way to display/analyze this data. The inter patient variation is causing the high error bars. May I suggest instead looking at the data from each patient sample and calculating the fold-change that occurs between T0 and T24, then graphing the distribution and SD of that fold change across like samples. Quantities should also be normalized to total number of bacteria rather than to per mL (16S can be used as a normalization tool). In this manner you get an actual idea of the number of copies per total number of bacteria.
e. Lastly, and perhaps most importantly, the way that I read this, these experiments were only run once with each patient sample counting for the n. This is incorrect. Each patient sample is actually a separate sample even though they are from a similar patient subset. Instead, each patient sample needs to be run in triplicate to ensure that the trends that were found are consistent. If I'm reading this incorrectly and these experiments were carried out 6-9 different times, you're good. If not, please repeat.
4. Supplementary Figure 2 makes a strong point and should be considered for the main text.
5. Grammatically, Line 24 = remove "Also"
Line 44 = change to "co-dependence, which is critical..."
Line 49 = "Obesity and overweight", for the purpose of this manuscript, aren't these the same thing? If not, I would expect four percentages to be described (two each for adults and children)
Line 146 = It is worth mentioning....
Comments on the Quality of English LanguageThere are minor errors in the manuscript that require editing, but overall, this article is well written.
Author Response
We thank the reviewer for the constructive and careful assessment of our manuscript. We have tried to address all the points raised (see attached document).

Round 2
Reviewer 3 Report
Comments and Suggestions for Authors
Thank you to the authors for making the suggested changes and including more explanation where necessary. These changes were sufficient to address my concerns, and helped to clarify the data.
Comments on the Quality of English LanguageThe English in this article is fine. A review for minor errors is all that is necessary.
Author Response
Dear Reviewer,
First at all, thank you very much for your effort and the assessment of our manuscript. It has, indeed, improved it and that is very much appreciated since, nowadays, this is not always the case.
We have conducted a careful proof-reading of the manuscript and introduced some modifications to improve the text understadability. We also identified some misspellings/errors that have been corrected as well.
These and other changes suggested by the editor can be seen in the new version of the manuscript (track changes mode).